# *Rhodococcus equi*-Derived Extracellular Vesicles Promoting Inflammatory Response in Macrophage through TLR2-NF-κB/MAPK Pathways

**DOI:** 10.3390/ijms23179742

**Published:** 2022-08-28

**Authors:** Zhaokun Xu, Xiujing Hao, Min Li, Haixia Luo

**Affiliations:** 1Life Science School, Ningxia University, Yinchuan 750021, China; 2Key Laboratory of Ministry of Education for Conservation and Utilization of Special Biological Resources in the Western, Ningxia University, Yinchuan 750021, China

**Keywords:** *Rhodococcus equi*, extracellular vesicles, virulence plasmid, macrophage, inflammatory response

## Abstract

*Rhodococcus equi* (*R. equi*) is a Gram-positive coccobacillus that causes pneumonia in foals of less than 3 months, which have the ability of replication in macrophages. The ability of *R. equi* persist in macrophages is dependent on the virulence plasmid pVAPA. Gram-positive extracellular vesicles (EVs) carry a variety of virulence factors and play an important role in pathogenic infection. There are few studies on *R. equi*-derived EVs (*R. equi*-EVs), and little knowledge regarding the mechanisms of how *R. equi*-EVs communicate with the host cell. In this study, we examine the properties of EVs produced by the virulence strain *R. equi* *103*^+^ (*103*^+^-EVs) and avirulenct strain *R. equi* *103^−^* (*103^−^*-EVs). We observed that *103*^+^-EVs and *103^−^*-EVs are similar to other Gram-positive extracellular vesicles, which range from 40 to 260 nm in diameter. The *103*^+^-EVs or *103^−^*-EVs could be taken up by mouse macrophage J774A.1 and cause macrophage cytotoxicity. Incubation of *103*^+^-EVs or *103^−^*-EVs with J774A.1 cells would result in increased expression levels of IL-1β, IL-6, and TNF-α. Moreover, the expression of TLR2, p-NF-κB, p-p38, and p-ERK were significantly increased in J774A.1 cells stimulated with *R. equi*-EVs. In addition, we presented that the level of inflammatory factors and expression of TLR2, p-NF-κB, p-p38, and p-ERK in J774A.1 cells showed a significant decreased when incubation with proteinase K pretreated-*R. equi*-EVs. Overall, our data indicate that *R. equi*-derived EVs are capable of mediating inflammatory responses in macrophages via TLR2-NF-κB/MAPK pathways, and *R. equi*-EVs proteins were responsible for TLR2-NF-κB/MAPK mediated inflammatory responses in macrophage. Our study is the first to reveal potential roles for *R. equi*-EVs in immune response in *R. equi*-host interactions and to compare the differences in macrophage inflammatory responses mediated by EVs derived from virulent strain *R. equi* and avirulent strain *R. equi*. The results of this study have improved our knowledge of the pathogenicity of *R. equi*.

## 1. Introduction

*Rhodococcus equi* (*R. equi*) is a Gram-positive bacterial widely dispersed in the environment. Young foals are susceptible to *R. equi*, the infected equine most commonly results in severe pneumonia, also tenosynovitis, ulcerative enterocolitis, and abdominal abscessation symptoms appear [1,2], which is a serious concern in the equine industry. *R. equi* is a facultative intracellular bacterium and mostly survives and multiplies inside its host macrophages [3,4]. Macrophage is an important component of innate and adaptive immunity, and represent the first line of defense upon infection of the host with the pathogen, which have the functions of phagocytosing and clearing pathogens, processing and presenting antigens, and releasing various cytokines [5]. Macrophages can eliminate intracellular bacterial pathogen via multiple mechanisms, including phagosome acidification, apoptosis, autophagy, and the production of oxygen and nitrogen components and cytokines, among other processes [6,7]. The pathogenicity of *R. equi* is derived from its ability to survive and replicate within macrophages of susceptible hosts [4]. The ability of *R. equi* that persist in macrophages is dependent on the virulence plasmid pVAPA, which was found in virulent strain of *R. equi* but not in the avirulent strain [8]. Virulence-associated protein A (VapA) is a membrane-active protein, which is encoded from pathogenic island of plasmid pVAPA and participates in the acidification inhibition of the *R. equi*-containing vacuole (RCV), play the central role in the replication of *R. equi* [9,10,11].

As the critical pattern recognition receptors (PRRs) on the host cells, Toll-like receptors (TLRs) are essential components of the innate immune response [12,13]. Engagement of TLRs by *R. equi* ligands is an early event in the interaction of *R. equi* with its host cell. The pathogen recognition through the TLRs gives rise to nuclear factor-κB (NF-κB) activation and a series of other cellular signaling events that result in the production of cytokines which can be a two-edged sword [12]. On one hand, the cytokines induced by TLRs are a necessary component of host defense, on the other hand, some cytokines are pro-inflammatory and result in host tissue damage. As one of the important TLRs, TLR2 plays a central role in innate immune responses to *R. equi*. It has been demonstrated that *R. equi* induces tumor necrosis factor α (TNF-α), interleukin-12 (IL-12), and NO through NF-κB activation mediated by TLR2, but not TLR4 [14]. The absence of TLR2 in vivo clearance of *R. equi* is compromised [14]. *R. equi* expresses some TLR2 ligands. Such as *R. equi* surface poprotein VapA, can activate TLR2 and induce inflammatory cytokines in macrophages [14]. In addition, Macrophages secrete multiple inflammatory factors in response to *R. equi* infection, such as interleukin-1β (IL-1β), interleukin-6 (IL-6), interferon-γ (IFN-γ), interferon-β (IFN-β), interleukin-10 (IL-10), and TNF-α in mouse or foal macrophages [15,16,17,18]. Different cytokines play different roles in the host defense against *R. equi*. IFN-γ, TNF-α, or IL-6 are required in host clearance of *R. equi*, whereas IL-1β or IL-10 were detrimental [19,20,21,22]. More interestingly, avirulent *R. equi* strains induced similar cytokines to virulent strain [15]. It means the knowledge about the mechanism of *R. equi*-induced inflammatory response is still limited.

Extracellular vesicles (EVs) are lipid bilayer vesicles, which are produced by almost all domains of life: bacteria, archaea, and eukaryotes [23,24]. The diameter of EVs from bacteria ranging from 30 to 500 nm, contains various bioactive cargos such as proteins, nucleic acids, and lipid [25]. Gram-positive bacterial EVs (also called membrane vesicles) have received relatively little attention in the literature, because the thick cell wall of Gram-positive bacteria makes it difficult for them to release EVs. Recently, the functions of EVs produced by Gram-positive bacteria have been described, which are involved in nutrient acquisition, stress response, delivery of virulence factors and invasion of host and immune regulation [26]. *R. equi* has also been shown to produce EVs, but there is no evidence for the role of *R. equi*-derived EVs (*R. equi*-EVs) during *R. equi* infect macrophages [27]. 

Given the lack of research about *R. equi*-EVs, this study aimed to characterize *R. equi*-EVs and investigate their role during macrophage infection. The findings will provide new insights into the pathogenic mechanisms employed by *R. equi* and open new opportunities for vaccine development.

## 2. Results

### 2.1. Characterization of EVs from R. equi (R. equi-EVs)

We investigated the virulence *R. equi* strain *103*^+^ (*103*^+^) and avirulent *R. equi 103^−^* strain (*103^−^*) release vesicles. Scanning electron microscope (SEM), Transmission electron microscopy (TEM), and dynamic light scattering (DSL) was used to characterize the shape and size of EVs derived from *103*^+^ (*103*^+^-EVs) or *103^−^* (*103^−^*-EVs). Visualization by SEM revealed that EVs with protruding spherical structure membranes were observed on the surface of *103*^+^ strain and *103^−^* strain (black arrowhead in Figure 1A). The ultracentrifugation precipitate of *103^−^* or *103*^+^ culture supernatants showed brown (Figure 1B). TEM further revealed that both *103*^+^ and *103^−^* produced EVs of spherical cup-shaped structures and were surrounded by a lipid bilayer (Figure 1C). The results of DLS showed that *103*^+^-EVs ranged from 40 to 260 nm and averaged 130 nm and *103^−^*-EVs ranged from 60 to 180 nm and averaged 105 nm (Figure 1D). There is no remarkable difference in the size distribution analysis confirmed that *103*^+^-EVs and *103^−^*-EVs (Figure 1E), although the averaged size of *103*^+^-EVs is slightly larger than that of *103^−^*-EVs. From the above results, we characterize the morphology of *103*^+^-EVs and *103^−^*-EVs, and there is no significant difference in the size distribution between *103*^+^-EVs and *103^−^*-EVs.

### 2.2. R. equi EVs Were Taken up by Macrophage

There is some evidence that bacteria’s EVs carry molecules involved in their interactions with their hosts [28]. To test whether *R. equi* derived-EVs can be taken by macrophage or not. Dio-labeled *103*^+^-EVs and *103^−^*-EVs (green signals) prior to co-incubation with macrophages for 1 h, both *103*^+^-EVs or *103^−^*-EVs were found in the cytoplasm of J77A4.1 cells (red signals), a green, fluorescent signal was not seen in control cells which were treated with Dio-labled PBS (Figure 2A). There is no significant difference in the average Dio signal strength analysis of the *103*^+^-EVs group or *103^−^*-EVs group (Figure 2B). Overall, we showed that both *103*^+^-EVs and *103^−^*-EVs can be up taken by macrophage.

### 2.3. R. equi-EVs Induced Cytotoxicity in J774A.1 Cells

Cell counting kit-8 assays were performed on J774A.1 cells in order to investigate the cytotoxic potential of *R. equi*-EVs in macrophage. J774A.1 cells were challenged at 12 h and 24 h with *103*^+^-EVs or *103^−^*-EVs in a dose-dependent manner. We verified that J774A.1 cells exposed to *103*^+^-EVs or *103^−^*-EVs within 12 h or 24 h showed decreased viability compared to the PBS-treated cells (Control). *103*^+^-EVs or *103^−^*-EVs concentrations between 1 μg/mL to 20 μg/mL significantly reduced the viability of macrophages in a dose-dependent manner over time. Furthermore, the viability of J774A.1 cells did not differ significantly between the *103^+^*-EVs and *103^−^*-EVs groups (Figure 3).

### 2.4. R. equi-EVs Activate Inflammatory Response in J774A.1 Cells

To evaluate the capacity of *R. equi*-EVs induced proinflammatory response, we analyzed the expression of inflammatory cytokine factor interleukin-1β (IL-1β), interleukin-6 (IL-6), tumor necrosis factor α (TNF-α), and interleukin-10 (IL-10) in J774A.1 cells with adding an increased dose of *103*^+^-EVs or *103^−^*-EVs (0.5 μg/mL and 5 μg/mL according to the protein concentration) for 6 and 24 h. The results showed that IL-1β, IL-6 and TNF-α were significantly increased in J774A.1 cells treated with *103*^+^-EVs and *103^−^*-EVs in a time- and dose-dependent manner (Figure 4A–C). The *103*^+^-EVs treatment did not stimulate the level of IL-10 at 6 h nor 24 h (Figure 4D). However, the expression of IL-10 was increased in J774A.1 cells with treatment of *103^−^*-EVs (5 μg/mL) at 24 h (Figure 4D). The observation suggested that both *103*^+^-EVs and *103^−^*-EVs could induce macrophage inflammatory, and *103^−^*-EVs might induce a higher inflammatory response in macrophages than *103*^+^-EVs.

### 2.5. R. equi-EVs Induced Inflammation Mediated by NF-κB/MAPK Pathways

The NF-κB and MAPK pathways are known to trigger inflammatory cytokine production by macrophages. NF-κB p65, a member of NF-κB, and the three main members of MAPKs are ERK, JNKs, and p38, which activate by phosphorylating [29]. To evaluate the effect of NF-κB and MAPK pathways in cytokines production induced by *R. equi*-EVs, the protein expression of phosphorylated NF-κB p56, ERK, JNK, and p38 were detected by western blotting after co-incubation of J774A.1 cells with *103*^+^-EVs or *103^−^*-EVs for 15 min, 30 min, 45 min, 60 min, and 120 min, with the group treated with PBS as the control. The results showed that compared to the control, induced *103*^+^-EVs or *103^−^*-EVs significantly increased phospho-NF-κB p65 (p-NF-κB p65), phospho-p38 (p-p38), phospho-ERK (p-ERK), and phospho-JNK (p-JNK) and reached its peak at 15 min (Figure 5A).

To research the regulation of IL-1β, IL-6, TNF-α, and IL-10 production by NF-κB, p38, ERK, and JNK signaling pathways, NF-κB inhibitor (BAY-117082), ERK inhibitor (PD98059), p38 inhibitor (SB203580), and JNK inhibitor (SP600125) were used to pre-treat J774A.1 cells for 2 h prior to incubation with *103*^+^-EVs or *103^−^*-EVs for 24 h. The expression levels of inflammatory cytokines IL-1β, IL-6, TNF-α and IL-10 were analyzed by ELISA. The expression level of IL-1β, IL-6 and TNF-α were significantly decreased in NF-Κb inhibited J774A.1 cells following *103*^+^-EVs or *103^−^*-EVs treatment compared with *103*^+^-EVs or *103^−^*-EVs-treated J774A.1 (Figure 5B,D). Inhibiting p38 with SB203580 or ERK with PD98059 in J774A.1 cells markedly decreased the expression of IL-1β, IL-6, while the level of TNF-α did not show significantly difference in *103*^+^-EVs or *103^−^*-EVs-treated J774A.1 cells compared with un-inhibited cell with *103*^+^-EVs or *103^−^*-EVs-treated J774A.1 (Figure 5B,D). While inhibiting NF-κB, p38, ERK or JNK did not stimulate the secretion of IL-10 in *103*^+^-EVs or*103^−^*-EVs-treated J774A.1 by comparing with PBS treatment (Control) (Figure 5E). Thus, taking these results together. We concluded that the robust inflammatory response induced by *R. equi*-EVs was largely dependent on the NF-κB and MAPKs signaling activation.

### 2.6. TLR2-NF-κB/MAPK Is Involved in R. equi-EVs Induced Macrophages Inflammatory Response

TLR2 plays a central role in innate immune responses to the intracellular bacterium *R. equi* [14]. TLR2 activates an inflammatory response through mediating NF-κB and MAPK signaling pathways [29]. To determine the roles of TLR2 in *R. equi*-EVs-induced macrophages inflammatory response, we measured the levels of TLR2 in J774A.1 cells treated with *103*^+^-EVs or *103^−^*-EVs in a time dependent manner. Western blot analysis demonstrated that in J774A.1 cells, TLR2 was significantly increased at 3 h, 6 h, 12 h and 24 h after *103*^+^-EVs or *103^−^*-EVs treatment and reached its peak at 6 h, respectively (Figure 6A). Then, TLR2 inhibitor C29 was used to pre-treat J774A.1 cells for 1 h at 37 °C after incubation with *103*^+^-EVs or *103^−^*-EVs for 24 h, the expression levels of inflammatory cytokines IL-1β, IL-6, TNF-α and IL-10 were detected by ELISA. The expression level of IL-1βand IL-6 were significantly decreased in TLR2 inhibited J774A.1 cells following *103*^+^-EVs or *103^−^*-EVs treatment compared with *103*^+^-EVs or *103^−^*-EVs-treated J774A.1 (Figure 6B,C). Moreover, the expression of IL-10 was significantly decreased in TLR2-inhibited J774A.1 cells following *103^−^*-EVs treatment compared with the *103^−^*-EVs group. While in TNF-α there was no significant difference in TLR2-inhibited J774A.1 cells following *103*^+^-EVs or *103^−^*-EVs treatment compared with *103*^+^-EVs or *103^−^*-EVs-treated J774A.1 (Figure 6D). In addition, inhibiting TLR2 with C29 did not stimulate the secretion of IL-10 in *103*^+^-EVs-treated J774A.1 in comparison to the PBS treatment (control) (Figure 6E). To further test whether the TLR2-induced inflammatory response is associated with NF-κB/MAPK pathways, we observed that the inhibition of TLR2 by C29 markedly suppressed the expression level of p-NF-κB p65, p-p38, p-ERK, and p-JNK in C29 treated J774A.1 cell following *103*^+^-EVs or *103^−^*-EVs incubation compared with *103*^+^-EVs or *103^−^*-EVs treated J774A.1 (Figure 6F–J). In summary, these data demonstrated that the inflammatory response induced by *R. equi*-EVs was dependent on the TLR2-NF-κB/MAPK pathways.

### 2.7. The Proteins Component of R. eui-EVs Is Essential for Inducing Inflammatory Response

Bacteria-derived EVs contain various bioactive proteins, which are known to be potent immune stimulators in the host environment [30]. In this study, our results indicated that *103*^+^-EVs or *103^−^*-EVs have a distinct protein profile compared to its parent strain (Figure 7A). It has demonstrated well that VapA is located at the outside of cell wall and is essential for *R. equi* replication in macrophages [8]. We used western blot analysis to show that VapA was present in *103*^+^-EVs but not in *103^−^*-EVs due to lack of virulence of the plasmid; while VapA cannot be detected when *103*^+^-EVs hydrolyze with Proteinase K (PK) (Figure 7B). In addition, in order to figure out the membrane associated proteins relevant with *R. equi*-EVs-induced inflammatory response, Proteinase K (PK) was used for the degradation of the proteins of *103*^+^-EVs and *103^−^*-EVs. As expected, treating *103*^+^-EVs and *103^−^*-EVs with PK significantly decreased the level of TLR2 compared with J774A.1incubated with *103*^+^-EVs or *103^−^-*EVs that were not hydrolyzed by PK. (Figure 7C). In addition, same trend can also be seen in the NF-κB/MAPK pathway (Figure 7D) and some inflammation factors, including IL-1β, IL-6, TNF-α, and IL-10 (Figure 7E–H). Taking these results together, we suggest that proteins exposed on the *R. equi*-EVs surface are mainly involved in the TLR2-NF-κB/MAPK-mediated inflammatory response in macrophage.

## 3. Discussion

*Rhodococcus equi* is a Gram-positive with complex mycolic acid cell wall, an intracellular bacterial pathogen, which can survive and replicate within alveolar macrophage in a phagosomal compartment that fails to mature into lysosome, resulting in the establishment of the *R. equi*-containing vacuole. Virulence-associated plasmids regulate the survival and multiplication of *R. equi* inside the macrophages. The plasmid-cured *R. equi* strains are unable survive in the macrophage and avirulent in foal and murine infection model [8]. In the present study, we represent the first study, to our knowledge, of the production of virulence strain *R. equi103*^+^ and plasmid-cured strain *R. equi 103^−^* EVs in cultural cultivation. The morphological characteristics of EVs from *R. equi* shared many common features with other Gram-positive bacterial [31,32,33]. Our data showed that *R. equi*-EVs were spherical with bilayer structures with a size range from 40~300 nm (Figure 1). Interestingly, we found that the averaged diameter of *103*^+^-EVs (130 nm) is slightly bigger than *103^−^*-EVs (105 nm) (Figure 1). The size of EVs in Gram-positive bacterial varies between different bacterial strains [34,35,36,37], throughout various stages of bacterial growth [38], and environmental conditions [27,39,40]. As the reports that the size distribution of EVs of the invasive Streptococcus pyogenes strains SSI-1 were 1.31 times bigger than the non-invasive Streptococcus pyogenes strains JRS4 [37]. The bigger size of EVs means it can carry more active molecules including virulence factor. Therefore, we speculate that the existence of virulence plasmid pVAPA might be the causee of the larger size of *103*^+^-EVs compared to *103^−^*-EVs. 

As of now, several studies indicate that Gram-positive EVs, including *Staphylococcus aureus* [41], *Streptococcus pneumoniae* [33], and *Filifactor alocis* [42] could induce macrophage activation and the production of an array of inflammatory cytokines. For instance, the expression levels of IL-1β and IL-18 increased significantly in macrophage which co-incubated with EVs derived from *Staphylococcus aureus* [41]. It is still unknown how EVs derived from *R. equi* affect macrophages. In our current report, we demonstrate that macrophages can phagocytoze EVs secreted by *R. equi* for the first time. We measured the function of *R. equi*-EVs in cellular inflammatory response. Our study revealed that *R.equi*-EVs could significantly enhance the production of IL-1β, IL-6, and TNF-α in J774A.1, which indicated *R. equi* as an intracellular pathogen that could effectively trigger cellular inflammatory response in macrophages through secretory vesicles. Both the 6 h and 24 h incubations of *103^+^*-EVs did not change the level of IL-10 expression (Figure 4D). Expression of IL-10 were only increased in J774A.1 cell with treatment of *103^−^*-EVs (5μg/mL) at 24 h (Figure 4D). The possible reason for the no change of IL-10 in *103*^+^-EVs-treated J77A4A.1 may be the anti-inflammatory reaction will be trigged after the 24 h treatment with *103*^+^-EVs.

An essential component of innate immunity against pathogens is the toll-like receptor family and its downstream signaling pathways. TLR2 is one of the important receptors for EVs produced by Gram-positive bacteria because the EVs contain various bioactive proteins which are known to be potent immune stimulators to the host cell [43,44]. Gram-positive bacteria have lipoproteins that are potent TLR2 stimulators. EVs released by *Mycobacterium tuberculosis* were enriched in lipoproteins, and intratracheal injection of these EVs in mice induced lung inflammation in a TLR2-dependent manner [45]. Hyun et al. found that EVs released by *Filifactor alocis* induced systemic bone loss through TLR2, and lipoprotein play an important role in EVs inducing systemic bone loss [46]. TLR2 play a central role in innate immune responses to the intracellular bacterium *R. equi*, while TLR4 was not involved in this response [14]. In this study, we demonstrated that *103*^+^-EVs or *103^−^*-EVs triggered an inflammatory response that was associated with TLR2-NF-κB/MAPK signaling pathways (Figure 3, Figure 4, Figure 5 and Figure 6). Above all, our results demonstrated that there is no significant difference between the *103*^+^-EVs and *103^−^-*EVs-induced inflammatory response, which indicated that many strains of *R. equi* have basic virulence potential, including the plasmid-less strain. As previously reported, sometimes sick foals yield plasmid-negative strains [47] and avirulent *R. equi* strains induced similar cytokines to virulent strains [15,18]. It suggests that chromosome and plasmid genes cooperate the pathogenicity and virulence and under certain circumstances and in certain hosts, plasmid-less strains may cause disease [8]. It has well demonstrated that the Virulence-associated protein A (VapA) is located on the cell surface and plays a role in TLR2 activation [14], which indicated that the vesicles surface might present VapA and regulate the activation of immune cells. The results in our study showed that VapA was presented in *103*^+^-EVs and as VapA can be degraded by Proteinase K (Figure 7B). Additionally, we observed EVs weakened their immune-stimulating effect when pretreated with protease. TLR2-NF-κB/MAPK signaling pathways could not be trigged after *R. equi*-EVs protein degradation. However, we do not know which proteins of *R. equi*-EVs play the main inflammatory function. Therefore, *R. equi*-EVs proteins composition needs to be further researched. 

In summary, we first characterized that EVs derived from virulence *R. equi 103*^+^ and avirulent *R. equi 103^−^* and demonstrated that *R. equi*-EVs significantly induced inflammatory responses via TLR2-NF-κB/MAPK pathway. We also demonstrated that proteins from *R. equi*-EVs trigged macrophage inflammatory response via TLR2-NF-κB/MAPK pathway (Figure 8). We found that *R. equi*-EVs are important in *R. equi* infection, which offered us a new perspective on *R. equi* infection mechanisms. 

## 4. Materials and Methods

### 4.1. Bacterial Strain and Culture Conditions

Virulent strain *R. equi 103*^+^(*103*^+^) and avirulent strain *R. equi 103^−^* (*103^−^*) were gifted by Wim G Merjer from University College Dublin. These strains were routinely grown with vigorous shaking at 37 °C in Brain-Heart Infusion Broth (BHI). All strains were stored at −80 °C in 80% BHI 20% glycerol (*vol*/*vol*).

### 4.2. Scanning Electron Microscope Detection of R. equi-EVs 

The *103*^+^ or *103^−^* strains were grown in BHI at 37 °C and harvested centrifugation. Bacterial cells were washed with PBS and fixed with 2.5% glutaraldehyde in PBS at 4 °C. The fixed bacterial cells were washed three times with PBS. Then, the sample was dehydrated with ethanol and further dehydrated by isopentyl acetate (Sigma-Aldrich, St. Louis, MO, USA). The samples were observed using cold field emission scanning electron microscope (SEM) (Hitachi, Tokyo, Japan).

### 4.3. Isolation of R. equi-EVs

EVs produced by *103*^+^ (*103*^+^-EVs) or *103^−^* (*103^−^*-EVs) were Isolation by ultracentrifugation. Briefly, sub-cultured *R. equi* were inoculated to BHI and grown at 37 °C with vigorous shaking to OD600 = 1.0. *R. equi* were pelleted at 12,000× *g* for 20 min, the supernatant was again at 16,000× *g* for 20 min to remove remaining cells. The supernatant was filtered through a 0.45 μm membrane (Millipore, Burlington, MA, USA), and concentrated 20-fold with 100-kDa cut-off centrifugal filter (Millipore, Burlington, MA, USA). The retentate was again filtered through a 0.22 μm membrane (Millipore, Burlington, MA, USA). The resulting filtrate was subjected to ultracentrifugation at 150,000× *g* for 3 h at 4 °C using a P70AT rotor (Hitachi, Tokyo, Japan). The pellet obtained was re-suspended in phosphate-buffered saline (PBS) and again subjected to ultracentrifugation at 150,000× *g* for 3 h at 4 °C. The precipitate was re-suspended in PBS. The protein concentrations of *R. equi*-EVs were quantified using the bicinchoninic acid (BCA) assay (Thermo Fisher Scientific Inc, Waltham, MA, USA). Finally, *R. equi*-EVs were stored at −80 °C until further characterization. Additionally, 10 μL of purified *R. equi*-EVs was grown in BHI agar plates to confirm that all *R. equi* cells were eliminated.

### 4.4. Transmission Electron Microscopy Detection of R. equi-EVs

The *103^+^*-EVs and *103^−^*-EVs were applied on grids which were negatively stained with 1% phosphotungstic acid for 1 min, followed by drying at room temperature. EVs were observed using a HT7700 transmission electron microscope (Hitachi, Tokyo, Japan) operated at 120 kV.

### 4.5. Dynamic Light Scattering Analysis

The size distribution of the *103*^+^-EVs or *103^−^*-EVs was confirmed by dynamic light scattering (DLS)measurements performed using a nanoparticle size and zeta potential analyzer (Anton Paar, Hohenbrugg, Austria). All the samples were prepared in PBS buffer pH 7.4. Then the samples were detected according to the operating instructions of the instrument.

### 4.6. Culture Conditions for J774A.1 Cells and R. equi-EVs Internalization Studies

The mouse macrophages cell line (ATCC#J774A.1) J774A.1 (Cellcook, Guangzhou, China) was cultivated in the DMEM medium, high glucose (Biological industries, Israel) supplemented with 10% fetal bovine serum (Biological Industries, Israel) at 37 °C in cell culture incubator containing 5% CO_2_. 

The *103*^+^-EVs or *103^−^*-EVs were stained with Dio (10 μM; Beyotime Biotechnology, Shanghai, China) for 20 min. The Dio-labeled *103*^+^-EVs or *103^−^*-EVs were centrifuged at 150,000× *g* for 3 h and washed in PBS. J774A.1 cells were plated onto a covered with sterile coverslips. Dio-labeled *103*^+^-EVs or *103^−^*-EVs were incubated with J774A.1 cells. The macrophages were fixed using 4% paraformaldehyde for 10 min. The macrophages were washed in PBS, and the nuclei were labeled with 4′,6-diamidino-2-phenylindole (Beyotime Biotechnology, Shanghai, China). A glass cover was added to the slide with neutral resin. Cells were stained with Rhodamine-labeled F-actin (Beyotime Biotechnology, Shanghai, China). The slides were visualized using a confocal microscope. 

### 4.7. Cytotoxicity Assay

For in vitro cytotoxicity assay, J774A.1 cells were seeded into a 96 well plate at 1 × 10^4^ per well at 37 °C in cell culture incubator containing 5% CO2. J774A.1 cells were treated for 12 h or 24 h with increasing protein concentrations of *103*^+^-EVs or *103^−^*-EVs (1 μg/mL~20 μg/mL). The Cell Counting Kit-8 (Beyotime Biotechnology, Shanghai, China) was used to measure the cytotoxicity of J774A.1 cells pretreated with or without *103*^+^-EVs or *103^−^*-EVs according to the manufacturer’s instructions.

### 4.8. Enzyme-Linked Immunosorbent Assay (ELISA)

For ELISA assay, J774A.1 cells were seeded into a 6 well plate at 1 × 10^6^ per well at 37 °C in cell culture incubator containing 5% CO_2_. For different experiments, J774A.1 cells were treated for 12 h or 24 h with increasing protein concentrations of *103^+^*-EVs or *103^−^*-EVs (1 μg/mL~20 μg/mL). In addition, the inhibitors of NF-κB (BAY-117082, 10 μM), p38 (SB203580, 10 μM), ERK (PD98059, 20 μM), JNK (SP600125, 20 μM), and TLR2 (C29, 100 μM) were used to pretreat for 2 h or 1 h before stimulation J774A.1 cells with *103^+^*-EVs or *103^−^*-EVs at a concentrations of 5 μg/mL for 24 h. Furthermore, J774A.1 cells were treated with native *103^+^*-EVs or native *103^−^*-EVs (5 μg/mL), proteinase K-treated (PK) *103^+^*-EVs or *103^−^*-EVs (5 μg/mL before proteinase K treatment) for 24 h respectively. The supernatants from different groups were collected. The protein levels of inflammatory cytokines interleukin 1beta (IL-1β), interleukin 6 (IL-6), tumor necrosis factor α (TNF-α), and interleukin 10 (IL-10) in the supernatants of macrophages stimulation were detected with ELISA Kits (BOSTER Biological Technology, Pleasanton, CA, USA).

### 4.9. SDS–PAGE Analysis of EVs and Total Bacteria Proteins

Total proteins and EVs extracted from *R. equi* were examined by sodium dodecyl sulphate–polyacrylamide gel electrophoresis (SDS–PAGE). Bacterial protein extraction kit (Beyotime Biotechnology, Shanghai, China) was used to prepare whole bacteria cell proteins. The protein concentrations of *R. equi*-EVs and whole bacteria cell proteins were quantified using the bicinchoninic acid (BCA) assay (Thermo Fisher Scientific Inc, Waltham, MA, USA). *R. equi*-EVs proteins and whole bacteria proteins were adjusted to a concentration of 15 μg per well on SDS-PAGE (4% polyacrylamide stacking gel, 12 % polyacrylamide separating gel). After electrophoresis, proteins were stained with coomassie brilliant blue (Beyotime Biotechnology, Shanghai, China) according to the supplier’s instructions.

### 4.10. Western Blotting

For western blotting, cell lysis buffer for western and IP (Bestbio, Shanghai, China) was used to prepare J774A.1 cell proteins. The protein concentrations of J774A.1 were quantified using the bicinchoninic acid (BCA) assay (Thermo Fisher Scientific Inc, Waltham, MA, USA). The samples of J774A.1 proteins was adjusted to a concentration of 30~50 μg perwell. After electrophoresis, protein samples were transferred to a polyvinylidene difluoride membrane that was blocked with 5% milk in TBS-T (TBS containing 0.05% Tween 20) for 1 h at room temperature. After washing the membrane with TBS-T, it was incubated with the specific antibodies against p-ERK (proteintech, Chicago, IL, USA), ERK (proteintech, USA), p-p38 (CST, Danvers, MA, USA), p38 (proteintech, Chicago, IL, USA), p-JNK (proteintech, Chicago, IL, USA), JNK (proteintech, Chicago, IL, USA), p-NF-κB p65 (CST, USA), NF-κB p65 (proteintech, Chicago, IL, USA), β-actin (proteintech, Chicago, IL, USA), or VapA (ABclonal, Wuhan, China) in dilution 1:1000 at 4 °C overnight. The membrane was washed with TBS-T and incubated with horseradish peroxidase-conjugated secondary antibodies for 1 h. Membrane was then washed thoroughly with TBS-T before the addition of a chemiluminescent substrate and exposure.

### 4.11. The Enzymatic Hydrolysis of EVs

To digest the EVs protein, 30 μg of *103*^+^-EVs or *103^−^*-EVs was treated with proteinase K (PK) (TransGen Biotech, Peking, China) for 30 min. The enzymes were inactivated at 75 °C for 1 h. J774A.1 cells were treated with native *103*^+^-EVs or native *103^−^*-EVs (5 μg/mL), PK-treated *103*^+^-EVs or *103^−^*-EVs (5 μg/mL before PK treatment) respectively.

### 4.12. Data Analysis

All data were analyzed by GraphPad Prism 7 software. The comparisons between groups were performed by one-way ANOVA followed by Tukey’s test. Each experiment was repeated at least three times. *p*-values > 0.05 were considered not statistically significant. *p*-values < 0.05 were considered statistically significant. * *p <* 0.05, ** *p* < 0.01, *** *p* < 0.001, and ns, not significant.

## Figures and Tables

**Figure 1 ijms-23-09742-f001:**
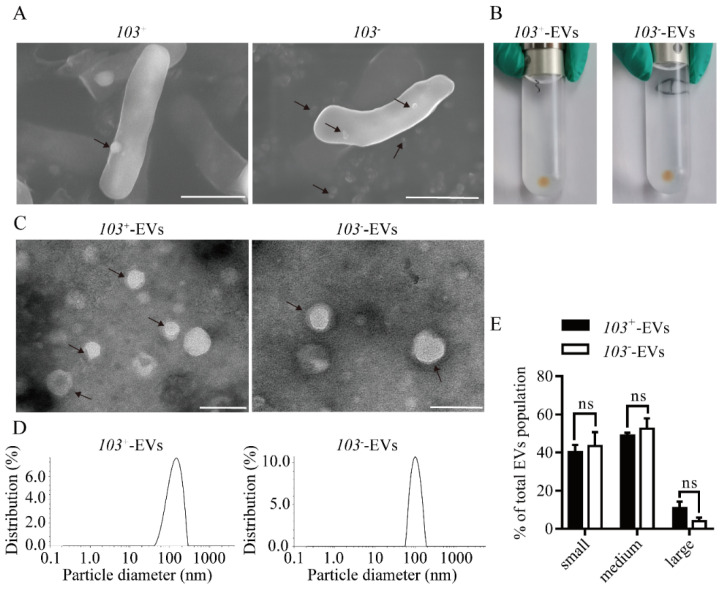
Characterization of extracellular vesicles (EVs) isolated from strains of *R. equi 103*^+^ (*103*^+^) and *R. equi 103^−^* (*103^−^*). (**A**) Scanning electron microscopy visualization of EVs production by strains of *103*^+^ or *103^−^*. Black arrow indicates EVs. Scale bars 1 um. (**B**) EVs pellet by ultracentrifugation from strains *103*^+^ (*103*^+^-EVs) or *103^−^* (*103^−^*-EVs). (**C**) *103*^+^-EVs or *103^−^*-EVs were visualized by negative staining transmission electron microscopy. Black arrow indicates EVs. Scale bars 100 nm. (**D**) Dynamic light scattering determination of size distribution for *103*^+^-EVs or *103^−^*-EVs, respectively. (**E**) Quantification by dynamic light scattering of the size distribution of small (<100 nm), medium (100–200 nm) and large (>200 nm) EVs produced by strains of *103*^+^ or *103^−^*, as indicated. The comparisons between groups were performed by one-way ANOVA followed by Tukey’s test. ns, not significant.

**Figure 2 ijms-23-09742-f002:**
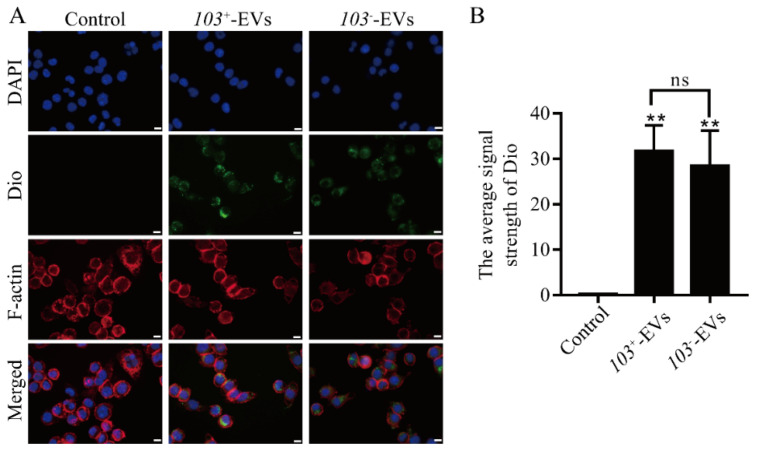
*103*^+^-EVs and *103^−^*-EVs are internalized by J774A.1 cells. (**A**) Representative confocal images of J774A.1 cells exposed to *103*^+^-EVs or *103^−^*-EVs for the indicated times. Green, Dio-stained *103*^+^-EVs or *103^−^*-EVs; Red, F-actin; blue, DAPI nuclear stain. The control group consisted of J774A.1 cells treated with Dio-PBS. Scale bars: 20 µm. (**B**) The average signal strength of Dio-labled *103*^+^-EVs or *103^−^*-EVs uptake by J774A.1 cells. The data were compared with the Dio-labled PBS stimulated control cells. The comparisons between groups were performed by one-way ANOVA followed by Tukey’s test. ** *p <* 0.01, and ns, not significant.

**Figure 3 ijms-23-09742-f003:**
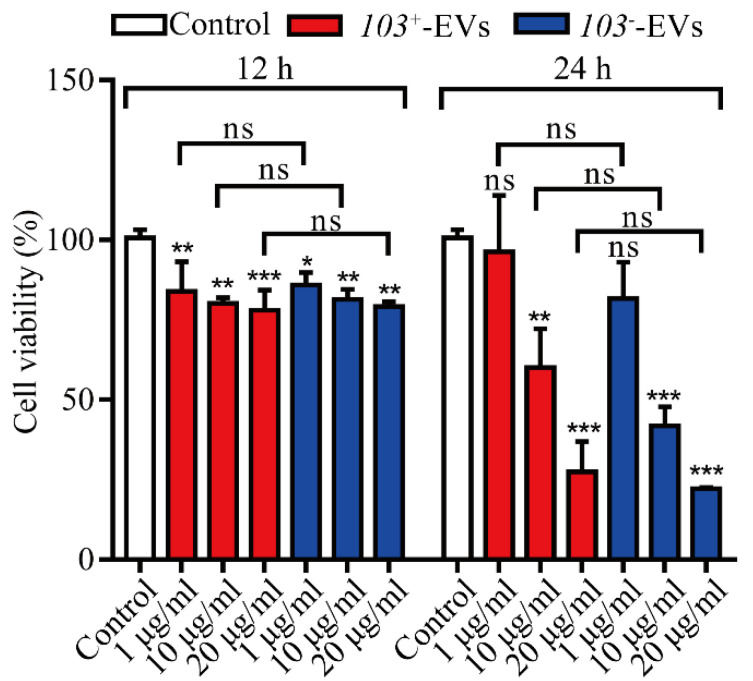
*103*^+^-EVs and *103^−^*-EVs induce cytotoxicity in J774A.1 cells. J774A.1 cells stimulated with various concentrations of *103*^+^-EVs and *103^−^*-EVs for 12 h or 24 h, then the cell viability measured by cell counting kit-8. The data were compared with the PBS stimulated control cells. The comparisons between groups were performed by one-way ANOVA followed by Tukey’s test. * *p <* 0.05, ** *p <* 0.01, *** *p <* 0.001, and ns, not significant.

**Figure 4 ijms-23-09742-f004:**
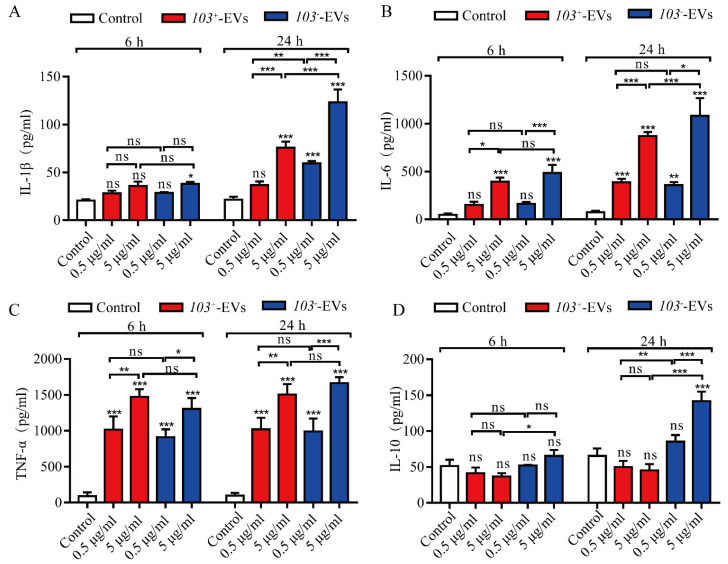
The *103*^+^-EVs and *103^−^*-EVs activate inflammatory responses in J774A.1 cells. The concentrations of (**A**) IL-1β, (**B**) IL-6, (**C**) TNF-α, and (**D**) IL-10 in supernatants from J774A.1 cells. The supernatants from the J774A.1 cells stimulated with various concentrations of *103*^+^-EVs or *103^−^*-EVs for 6 h or 24 h were collected, and the inflammatory cytokines levels were measured by ELISA. The data were compared with the PBS stimulated control cells. The comparisons between groups were performed by one-way ANOVA followed by Tukey’s test. * *p <* 0.05, ** *p <* 0.01, *** *p <* 0.001, and ns, not significant.

**Figure 5 ijms-23-09742-f005:**
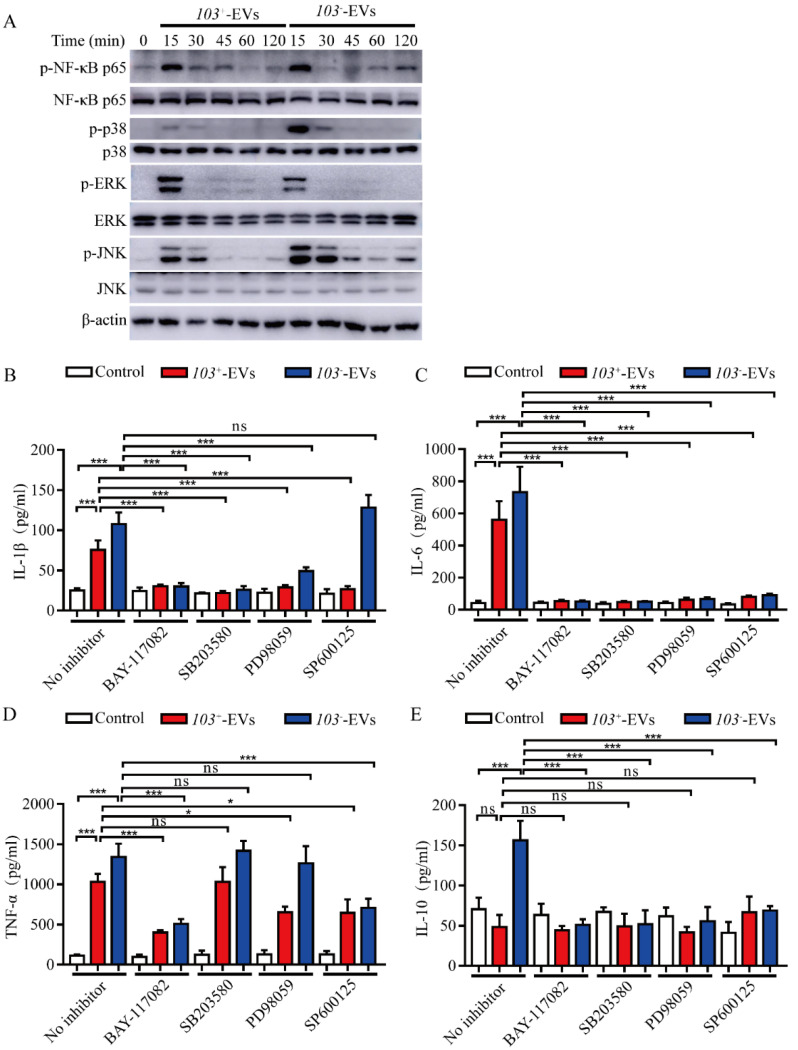
The *103*^+^-EVs and *103^−^*-EVs induced inflammatory cytokines secretion was regulated by NF-κB, and MAPK signaling pathways. (**A**) *103*^+^-EVs and *103^−^*-EVs induced the phosphorylation of NF-κB p65, p38, ERK and JNK protein in J774A.1 cells. J774A.1 cells were co-incubated with *103*^+^-EVs or *103^−^*-EVs at concentrations of 5 μg/mL for 0 min (control), 15 min, 30 min, 45 min, 60 min, and 120 min, the phosphorylated and total protein of NF-κB p65, p38, ERK and JNK were detected by western blotting. The concentrations of (**B**) IL-1β, (**C**) IL-6, (**D**) TNF-α, and (**E**) IL-10 in the culture supernatants. The inhibitors of NF-κB (BAY-117082, 10 μM), p38 (SB203580, 10 μM), ERK (PD98059, 20 μM), and JNK (SP600125, 20 μM) were used to pretreat for 2 h before stimulation J774A.1 cells with *103*^+^-EVs or *103^−^*-EVs at concentrations of 5 μg/mL for 24 h, and then collected supernatants. The data were compared with the no inhibitor stimulated control cells. The comparisons between groups were performed by one-way ANOVA followed by Tukey’s test. * *p <* 0.05, *** *p <* 0.001, and ns, not significant.

**Figure 6 ijms-23-09742-f006:**
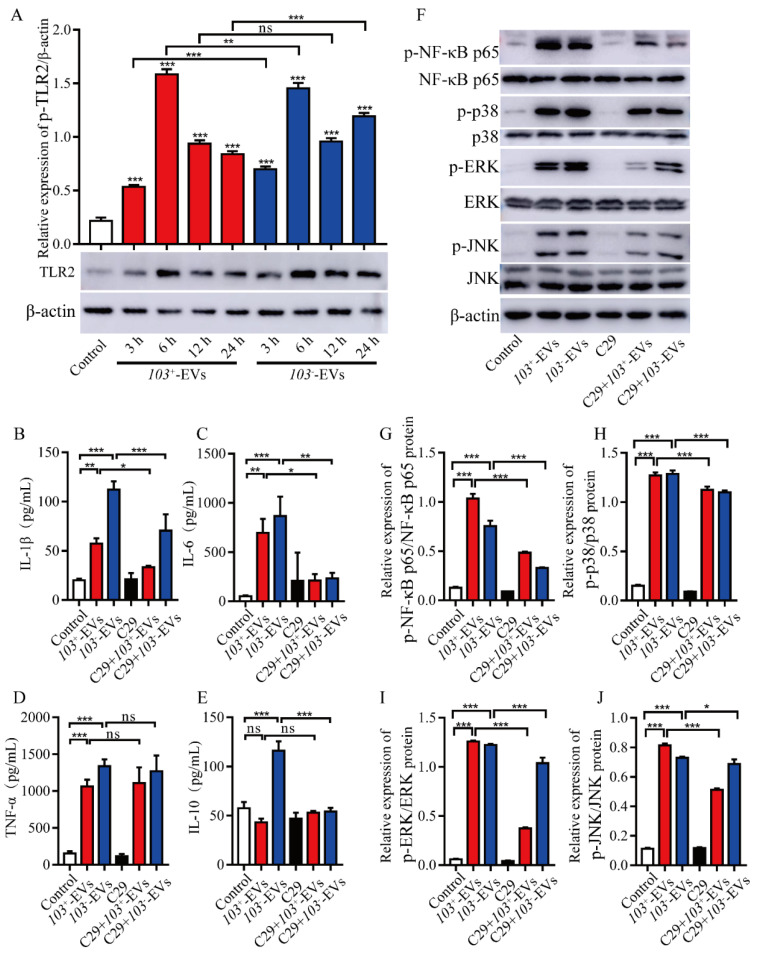
*103*^+^-EVs and *103^−^*-EVs induced cytokines secretion was regulated by TLR2 signaling pathway. (**A**) J774A.1 cells were co-incubated with *103*^+^-EVs or *103^−^*-EVs at concentrations of 5 μg/mL for 3 h, 6 h, 12 h, and 24 h, the protein of TLR2 was detected by western blotting. The results of western blotting were analyzed by gray scale. Inhibitors of TLR2 (C29, 100 μM) was used to pretreat for 1 h before stimulation J774A.1 cells with *103*^+^-EVs or *103^−^*-EVs at concentrations of 5 μg/mL for 24 h, and then detected the protein levels of (**B**) IL-1β, (**C**) IL-6, (**D**) TNF-α, and (**E**) IL-10 in the culture supernatants by ELISA. (**F**) Inhibitors of TLR2 (C29, 100μM) was used to pretreat for 1 h before stimulation J774A.1 cells with *103*^+^-EVs or *103^−^*-EVs at concentrations of 5 μg/mL for 15 min, and then detected the phosphorylation of NF-κB p65, p38, ERK and JNK protein by western blotting. The results of western blotting were analyzed by gray scale. Relative gray value shows the ratio of (**G**) Phospho-NF-κB p65 (p-NF-κB p65) to total NF-κB p65 protein, (**H**) Phospho-p38 (p38) to total p38 protein, (**I**) Phospho-ERK (p-ERK) to total ERK protein, and (**J**) Phospho-JNK (p-JNK) to total JNK protein. The data were compared with the control cells. The comparisons between groups were performed by one-way ANOVA followed by Tukey’s test. * *p <* 0.05, ** *p <* 0.01, *** *p <* 0.001, and ns, not significant.

**Figure 7 ijms-23-09742-f007:**
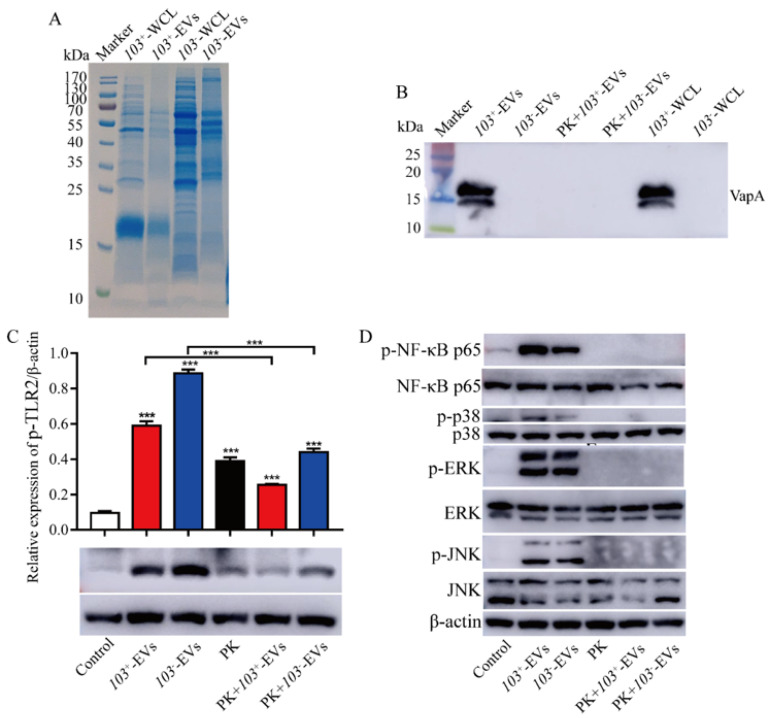
The reduced content of *103*^+^-EVs and *103^−^-*EVs proteins decreased the inflammatory effects induced by *103*^+^-EVs or *103^−^-*EVs. (**A**) SDS-PAGE analysis of the *103*^+^-EVs and *103^−^-*EVs proteins. WCL: whole bacteria cell lyste. (**B**) Western blotting analysis of the exist of virulence associated protein A in *103*^+^-EVs and *103^−^-*EVs. PK: proteinase K-treated. (**C**,**D**) J774A.1 cells were treated with native *103*^+^-EVs or native *103^−^-*EVs (5 μg/mL), proteinase K-treated *103*^+^-EVs or *103^−^-*EVs (5 μg/mL before proteinase K treatment) for different times, respectively, and then detected the (**C**) TLR2 protein, (**D**) phosphorylation of NF-κB p65, p38, ERK and JNK protein by western blotting. The concentrations of (**E**) IL-1β, (**F**) IL-6, (**G**) TNF-α, and (**H**) IL-10 in supernatants from J774A.1 cells. J774A.1 cells were treated with native *103*^+^-EVs or native *103^−^*-EVs (5 μg/mL), proteinase K-treated (PK) *103*^+^-EVs or *103^−^*-EVs (5 μg/mL before proteinase K treatment) for 24 h, respectively, and then collected the supernatants. The concentrations of inflammatory factor measured by ELISA. The data were compared with the control cells. The comparisons between groups were performed by one-way ANOVA followed by Tukey’s test. * *p <* 0.05, ** *p <* 0.01, *** *p <* 0.001, and ns, not significant.

**Figure 8 ijms-23-09742-f008:**
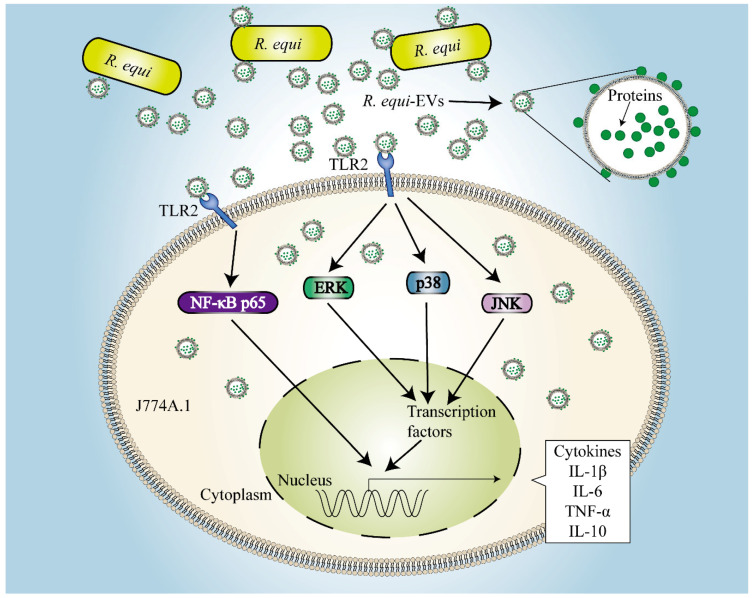
The model of *R. equi*-derived extracellular vesicles (*R. equi*-EVs) promoting inflammatory response in macrophage through TLR2-NF-κB/MAPK pathways.

## Data Availability

The original contributions presented in the study are included in the article.

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
