# Peer review of "Rhodococcus equi*-Derived Extracellular Vesicles Promoting Inflammatory Response in Macrophage through TLR2-NF-κB/MAPK Pathways"

_ijms, 2022, doi:10.3390/ijms23179742_

Round 1

Reviewer 1 Report

      The paper entitled Rhodococcus equi - derived Extracellular Vesicles Promoting Ininflammatory Response in Macrophage through TLR2-NF- 3 κB/MAPK pathways characterizes R. equi - EVs and investigates their role during macrophage infection. 

     In this paper, the Authors examined the properties of EVs produced by virulence strain R. equi 103+ (103+-EVs) and avirulent strain R. equi 103- (103- -EVs). They observed that 103+-EVs and 103- -EVs are similar to other Gram-positive extracellular vesicles, which range from 40 to 260 nm in diameter. The 103+-EVs or 103- -EVs were taken up by mouse macrophages of J774A.1 cell line and cause its cytotoxicity by increased expression of cytokines: IL-1β, IL-6, and TNF-α Additionally, the expression of TLR2, p-NF-κB, p-p38, and p-ERK were significantly increased in J774A.1 cells stimulated with R. equi-EVs. Also, the level of inflammatory factors and expression of TLR2, p-NF-κB, p-p38, and p-ERK in J774A.1 cells showed a significant decrease when incubated with proteinase K pretreated-R. equi-EVs.

The title of the paper is brief and informative. In the Introduction, the importance of the topic discussed in this paper was marked, especially in the context of the role of and the impact of Rhodococcus equi (R. equi) on macrophage activity. Because the Authors studied the impact of extracellular vesicles of R. equi on macrophage function, I recommend adding some information describing the role of macrophages in the regulation of the immune response and their key functions. The results presented in the paper are described very precisely.  Also, the figure’s legends contain a comprehensive description, but the information about statistical significance should contain the information about the reference sample. Discussion is very illustrative, well-supported by literature, the obtained results are summarized, and finally, a potential mechanism of pro-inflammatory activity of extracellular vesicles derived from Rhodococcus equi was proposed.

    Despite the interesting and actual subject presented in this paper, the particular sections of Materials and Methods must be completed. The number of cells used in point 4.7 should be added. A precise procedure of stimulation of  J774A.1 cells in point 4.8 must be added. Also exact conditions of SDS PAGE analysis (point 4.9) and Western blotting (point 4.10)(the protein concentration detection method, the amount of protein per well, antibodies dilutions, etc).

I recommended that a major revision is needed. 

Author Response

Response to Reviewer 1 Comments

Dear Reviewer,

Thank you for your time and  your comments for our manuscript of“Rhodococcus equi-derived Extracellular Vesicles Promoting Inflammatory Response in Macrophage through TLR2-NF-κB/MAPK pathways” (Manuscript ID ijms-1870601). We have taken all these comments and suggestions into account, and have made modifies in the uploaded  revised manuscript for your review.

Comments:

The paper entitled Rhodococcus equi - derived Extracellular Vesicles Promoting Ininflammatory Response in Macrophage through TLR2-NF-κB/MAPK pathways characterizes R. equi-EVs and investigates their role during macrophage infection.

 In this paper, the Authors examined the properties of EVs produced by virulence strain R. equi 103+ (103+-EVs) and avirulent strain R. equi 103- (103--EVs). They observed that 103+-EVs and 103- -EVs are similar to other Gram-positive extracellular vesicles, which range from 40 to 260 nm in diameter. The 103+-EVs or 103--EVs were taken up by mouse macrophages of J774A.1 cell line and cause its cytotoxicity by increased expression of cytokines: IL-1β, IL-6, and TNF-α. Additionally, the expression of TLR2, p-NF-κB, p-p38, and p-ERK were significantly increased in J774A.1 cells stimulated with R. equi-EVs. Also, the level of inflammatory factors and expression of TLR2, p-NF-κB, p-p38, and p-ERK in J774A.1 cells showed a significant decrease when incubated with proteinase K pretreated-R. equi-EVs.

The title of the paper is brief and informative. In the Introduction, the importance of the topic discussed in this paper was marked, especially in the context of the role of and the impact of Rhodococcus equi (R. equi) on macrophage activity. Because the Authors studied the impact of extracellular vesicles of R. equi on macrophage function, I recommend adding some information describing the role of macrophages in the regulation of the immune response and their key functions. The results presented in the paper are described very precisely. Also, the figure’s legends contain a comprehensive description, but the information about statistical significance should contain the information about the reference sample. Discussion is very illustrative, well-supported by literature, the obtained results are summarized, and finally, a potential mechanism of pro-inflammatory activity of extracellular vesicles derived from Rhodococcus equi was proposed.

Despite the interesting and actual subject presented in this paper, the particular sections of Materials and Methods must be completed. The number of cells used in point 4.7 should be added. A precise procedure of stimulation of J774A.1 cells in point 4.8 must be added. Also exact conditions of SDS PAGE analysis (point 4.9) and Western blotting (point 4.10)(the protein concentration detection method, the amount of protein per well, antibodies dilutions, etc).

We also appreciate your clear and detailed feedback and hope that the explanation has fully addressed all of your concerns. In the remainder of this letter, we discuss each of your comments individually along with our corresponding responses.

Point 1: Because the Authors studied the impact of extracellular vesicles of R. equi on macrophage function, I recommend adding some information describing the role of macrophages in the regulation of the immune response and their key functions. 

Response 1: We supplemented some of the content by reviewing the literature to reflect that the role of macrophages in the regulation of the immune response in R.equi infection. The following is the modified content, and the part marked in yellow is the supplemented content (lines 41-49 of introduction).

Point 2: Also, the figure’s legends contain a comprehensive description, but the information about statistical significance should contain the information about the reference sample.

Response 2: We supplemented the information about statistical significance and reference sample in all of figure’s legends. The modified content marked in yellow is the supplemented content (lines 119-120, 138-140, 154-155, 174-175, 214-215, 255-256, and 286-287 of figure’s legends).

Point 3: Despite the interesting and actual subject presented in this paper, the particular sections of Materials and Methods must be completed. The number of cells used in point 4.7 should be added. A precise procedure of stimulation of J774A.1 cells in point 4.8 must be added. Also exact conditions of SDS PAGE analysis (point 4.9) and Western blotting (point 4.10)(the protein concentration detection method, the amount of protein per well, antibodies dilutions, etc).

Response 3: We have revised the content of the materials and methods section. Modifications include: Added information on the number of J774A.1 cells used in point 4.7 (lines 417-418 of materials and methods). Re-described all experimentthe precise procedure of stimulation of J774A.1 cells in point 4.8 (lines 424-433 of materials and methods). The exact conditions of SDS-PAGE (point 4.9) and Western blotting (point 4.10) include the protein concentration detection method, the amount of protein per well, and antibodies dilutions, etc were added in revised manuscript and marked in yellow (lines 438-458 of materials and methods).

We would like to take this opportunity to thank you for all your time involved and this great opportunity for us to improve the manuscript. We hope you will find this revised version satisfactory.

Sincerely,

Zhaokun Xu

Reviewer 2 Report

Dear Authors,

The manuscript "Rhodococcus equi-derived Extracellular Vesicles Promoting Inflammatory Response in Macrophage through TLR2-NF-κB / MAPK pathways" presented to me for evaluation raises important questions about the pathogenicity of Rhodococcus equi. Weighing the pathogen of domestic animals. The frequency of infection with this bacterium can reach nearly 60%. Therefore, the research presented in the manuscript is original and has great cognitive value, because we know little about it so far. This state of knowledge creates a wide field for both Authors and future researchers to search.

Major revision

1. Poor introduction.

It is absolutely necessary to improve the introduction taking into account the current state of knowledge about the immune response / inflamatory response to Rhodococcus equi infection. There is little research in this area, so the Authors should have no problem to enrich the introduction with them, which will definitely affect the value of the entire study.

Minor Revision

1. Literature poor, but specific to the topic. Supplement with publications on the immune response to R.equi. infections

2. Literature error (Line 287). "Those results consistent with Kristine et al. Reported that chromosome and plasmid genes cooperate the pathogenicity and virulence and under certain circumstances and in certain hosts, plasmid less strains can cause life-threatening disease [4]". Kristine et al. it is not [4]. Correction needed.

Author Response

Response to Reviewer 2 Comments

Dear Reviewer,

Thank you for your time and  your comments for our manuscript of“Rhodococcus equi-derived Extracellular Vesicles Promoting Inflammatory Response in Macrophage through TLR2-NF-κB/MAPK pathways” (Manuscript ID ijms-1870601). We have taken all these comments and suggestions into account, and have made modifies in the uploaded  revised manuscript for your review.

Comments:

The manuscript "Rhodococcus equi-derived Extracellular Vesicles Promoting Inflammatory Response in Macrophage through TLR2-NF-κB / MAPK pathways" presented to me for evaluation raises important questions about the pathogenicity of Rhodococcus equi. Weighing the pathogen of domestic animals. The frequency of infection with this bacterium can reach nearly 60%. Therefore, the research presented in the manuscript is original and has great cognitive value, because we know little about it so far. This state of knowledge creates a wide field for both Authors and future researchers to search.

Major revision

  1. Poor introduction.

It is absolutely necessary to improve the introduction taking into account the current state of knowledge about the immune response / inflamatory response to Rhodococcus equi infection. There is little research in this area, so the Authors should have no problem to enrich the introduction with them, which will definitely affect the value of the entire study.

Minor Revision

  1. Literature poor, but specific to the topic. Supplement with publications on the immune response to R.equi. infections

  1. Literature error (Line 287). "Those results consistent with Kristine et al. Reported that chromosome and plasmid genes cooperate the pathogenicity and virulence and under certain circumstances and in certain hosts, plasmid less strains can cause life-threatening disease [4]". Kristine et al. it is not [4]. Correction needed.

We also appreciate your clear and detailed feedback and hope that the explanation has fully addressed all of your concerns. In the remainder of this letter, we discuss each of your comments individually along with our corresponding responses.

Point 1: It is absolutely necessary to improve the introduction taking into account the current state of knowledge about the immune response / inflamatory response to Rhodococcus equi infection. There is little research in this area, so the Authors should have no problem to enrich the introduction with them, which will definitely affect the value of the entire study.

Response 1: As suggested, we supplemented some of the content by reviewing the literature to reflect that the immune response / inflamatory response in R.equi infection. The following is the modified content, and the part marked in yellow is the supplemented content (lines 41-77 of introduction).

Point 2: Literature poor, but specific to the topic. Supplement with publications on the immune response to R.equi. infections

Response 2: We have supplemented some representative literature on immune/inflammatory responses in R.equi infection that are relevant to this article (lines 493-498 and 512-534 of references). The main relevant content of these references is reflected in the introduction (lines 41-77 of introduction).

Point 3: Literature error (Line 287). "Those results consistent with Kristine et al. Reported that chromosome and plasmid genes cooperate the pathogenicity and virulence and under certain circumstances and in certain hosts, plasmid less strains can cause life-threatening disease [4]". Kristine et al. it is not [4]. Correction needed.

Response 3: We have carefully and thoroughly proofread the manuscript to correct all the literature errors. The content of the original manuscript here is derived from the review (von Bargen K, Haas A. Molecular and infection biology of the horse pathogen Rhodococcus equi. FEMS Microbiol Rev. 2009 Sep;33(5):870-91.), and our description of this section may not be entirely appropriate. We have rewritten the content here and updated the cited references (lines 341-345 of discussion).

We would like to take this opportunity to thank you for all your time involved and this great opportunity for us to improve the manuscript. We hope you will find this revised version satisfactory.

Sincerely,

Zhaokun Xu

Reviewer 3 Report

Manuscript ID: ijms- 1870601

Type of manuscript: article

Title:  Rhodococcus equi-derived Extracellular Vesicles Promoting Inflammatory
Response in Macrophage through TLR2-NF-
κB/MAPK pathways

Journal: International Journal of Molecular Sciences

In this article, the authors investigated the properties of extracellular vesicles (EVs) produced by virulence strain R. equi 103+ (103+-EVs) and avirulenct strain R. equi 103- (103- -EVs). They observed that 103+-EVs and 103- -EVs are similar to other Gram-positive extracellular vesicles. The incubation of 103+-EVs or 103- -EVs with J774A.1 cells result in increased expression levels of IL-1β, IL-6, and TNF-α and the expression 21 of TLR2, p-NF-κB, p-p38, and p-ERK were significantly increased in J774A.

The paper is well organized and well written and represents a contribution in the field of the pathogenic mechanisms employed by R. equi and opening new opportunities for vaccine development. I personally recommend the publication of this paper.

I have only a suggestion: the authors could be to explain better, in the introduction or in the discussion, the role of TLR2 in the infection by R.equi.

Author Response

Response to Reviewer 3 Comments

Dear Reviewer,

Thank you for your time and  your comments for our manuscript of“Rhodococcus equi-derived Extracellular Vesicles Promoting Inflammatory Response in Macrophage through TLR2-NF-κB/MAPK pathways” (Manuscript ID ijms-1870601). We have taken all these comments and suggestions into account, and have made modifies in the uploaded  revised manuscript for your review.

Comments:

In this article, the authors investigated the properties of extracellular vesicles (EVs) produced by virulence strain R. equi 103+ (103+-EVs) and avirulenct strain R. equi 103- (103- -EVs). They observed that 103+-EVs and 103- -EVs are similar to other Gram-positive extracellular vesicles. The incubation of 103+-EVs or 103- -EVs with J774A.1 cells result in increased expression levels of IL-1β, IL-6, and TNF-α and the expression of TLR2, p-NF-κB, p-p38, and p-ERK were significantly increased in J774A.1.

The paper is well organized and well written and represents a contribution in the field of the pathogenic mechanisms employed by R. equi and opening new opportunities for vaccine development. I personally recommend the publication of this paper.

I have only a suggestion: the authors could be to explain better, in the introduction or in the discussion, the role of TLR2 in the infection by R.equi.

We also appreciate your clear and detailed feedback and hope that the explanation has fully addressed all of your concerns. In the remainder of this letter, we discuss your comments with our corresponding responses.

Point 1: I have only a suggestion: the authors could be to explain better, in the introduction or in the discussion, the role of TLR2 in the infection by R.equi.

Response 1: Thank you for your suggestion. As suggested, we supplemented some of the content by reviewing the literature to reflect that the role of TLR2 in the infection by R.equi. The following is the modified content, and the part marked in yellow is the supplemented content (lines 56-77 of introduction).

We would like to take this opportunity to thank you for all your time involved and this great opportunity for us to improve the manuscript. We hope you will find this revised version satisfactory.

Sincerely,

Zhaokun Xu
